# Using passive sensor data to probe associations of social structure with changes in personality: A synthesis of network analysis and machine learning

**Damien Lekkas**[1,2]*, **Joseph A. Gyorda**[1,2], **Erika L. Moen**[2,3], **Nicholas C. Jacobson**[1,2,3,4]

**1** Center for Technology and Behavioral Health, Geisel School of Medicine, Dartmouth College, Lebanon, New Hampshire, United States of America, **2** Quantitative Biomedical Sciences Program, Dartmouth College, Hanover, New Hampshire, United States of America, **3** Department of Biomedical Data Science, Geisel School of Medicine, Dartmouth College, Lebanon, New Hampshire, United States of America, **4** Department of Psychiatry, Geisel School of Medicine, Dartmouth College, Hanover, New Hampshire, United States of America

* Damien.Lekkas.GR@dartmouth.edu

**Data Availability Statement:** All relevant data are within the article and its Supporting Information files.

## Abstract

Social network analysis (SNA) is an increasingly popular and effective tool for modeling psychological phenomena. Through application to the personality literature, social networks, in conjunction with passive, non-invasive sensing technologies, have begun to offer powerful insight into personality state variability. Resultant constructions of social networks can be utilized alongside machine learning-based frameworks to uniquely model personality states. Accordingly, this work leverages data from a previously published study to combine passively collected wearable sensor information on face-to-face, workplace social interactions with ecological momentary assessments of personality state. Data from 54 individuals across six weeks was used to explore the relative importance of 26 unique structural and nodal social network features in predicting individual changes in each of the Big Five (5F) personality states. Changes in personality state were operationalized by calculating the weekly root mean square of successive differences (RMSSD) in 5F state scores measured daily via self-report. Using only SNA-derived features from wearable sensor data, boosted tree-based machine learning models explained, on average, approximately 28–30% of the variance in individual personality state change. Model introspection implicated egocentric features as the most influential predictors across 5F-specific models, with network efficiency, constraint, and effective size measures among the most important. Feature importance profiles for each 5F model partially echoed previous empirical findings. Results support future efforts focusing on egocentric components of SNA and suggest particular investment in exploring efficiency measures to model personality fluctuations within the workplace setting.

**Funding:** DL, JAG, and NCJ received funding through a National Institute on Drug Abuse (NIDA; https://nida.nih.gov/) institutional grant (5P30DA02992610). The funders had no role in study design, data collection and analysis, decision to publish, or preparation of the manuscript.

**Competing interests:** The authors have declared that no competing interests exist.

## 1. Introduction

Personality can be broadly defined as an individual's recurring behaviors and dispositions [1]; however, more nuanced perspectives exist in the psychological literature. One classical view frames personality as static representations of the individual. Indeed, this person-based perspective defines personality by traits that represent stable dispositions, highlighting interindividual rather than intraindividual variation. A popular model that evolved from this framework is the Five Factor (5F) model [2, 3]. This model operationalizes personality in terms of five primary traits: extraversion/introversion, agreeableness/antagonism, conscientiousness/disinhibition, stability/neuroticism, and openness/close-mindedness. The 5F Model has been applied to a variety of psychological constructs and contexts, including anxiety [4], depression [5], eating disorders [6, 7], mental health treatment outcomes [8], academic performance [9], and organizational dynamics [10]. Moreover, research that has applied the 5F model within the mental health domain has frequently shown phenomenologically informative interactions among the five trait dimensions [5, 8, 11].

Despite the prevalence of the 5F model and its trait-based perspective of personality, it is noteworthy that this model is not designed to consider momentary, within-person fluctuations of personality [12]. Accordingly, a competing view within the literature frames personality as a state and emphasizes a situation-based perspective to personality expression. Under this paradigm, personality states represent dynamic dispositions that vary by social and environmental context and thus emphasize analysis of intraindividual differences [13]. Overall, the dichotomy between trait- and state-based personality stems principally from biological (of the former) and social-cognitive (of the latter) theoretical foundations that differentially emphasize the impact of nature and nurture on behavioral phenotypes.

Although the person- and situation-based perspectives of personality have traditionally been in competing opposition, recent literature has suggested that they may be integrated to good effect. In one study, researchers leveraged thousands of ecological momentary assessments (EMAs) to reveal that individuals may visit certain environments more frequently in accordance with their stable 5F personality traits [14]. The results suggested that an individual's environment may positively reinforce (i.e., lower the variability of) their dominant personality attributes over time and result in an increased stability of specific personality dimensions. In fact, more recent personality theories have begun to reconcile both the person-perspective and situation-perspective [15, 16]. This reconciliation falls under the umbrella of Whole Trait Theory (WTT). WTT posits that individuals will experience each of the 5F traits throughout their lives, but to varying degrees from day-to-day [17]. As WTT has become an increasingly popular approach to operationalize personality in the recent literature [18], there is strong empirical precedent to build upon existing efforts in this direction. One manner with which to accomplish this is through creative methods development, specifically the application of quantitative techniques that have been proven to be useful in related psychological domains and provide an intuitive framework to model and explore state-trait personality duality in uniquely insightful ways.

One such suite of analytic techniques, social network analysis (SNA), has become an increasingly popular and effective tool for modeling and investigating psychological phenomena [19], notably personality and agency, which oftentimes fall within the purview of organizational psychology [20, 21]. A social network is a representation of the relationships between social actors or nodes, which can be individuals, groups, and organizations [22]. Analyses of social networks focus on the structure of relationships—represented as edges, or ties—in the network, and particularly what might facilitate or restrict the exchange of information. Importantly, social networks allow for the examination of social processes over time in order to

reveal how network ties evolve and actors are influenced [23, 24]. Given that individuals' personalities likely influence (and are influenced by) those with whom they interact [14], it is reasonable to apply social networks—models of interpersonal relationships—to study personality dynamics. While the connection between social networks and personality has been established in the literature, the majority of works that examine personality within a social network context do so by modeling personality as a node attribute to examine its influence on tie formation and other network processes [25, 26]. Operationalizing personality in this manner is inherently a trait-based framing—information on static measures of personality complements the social processes made observable and quantifiable through SNA. In light of WTT and other theories that view personality as fluid and determinable via intrapersonal and interpersonal factors, changes to personality state can also be interrogated as an outcome of the network structure itself. Therefore, the interest of the present analysis is to leverage SNA as a means of exploring social structural attributes that are most predictive of personality state change.

Along with SNA, machine learning presents itself as a powerful analytical tool within the personality literature and has shown success in predicting personality types and traits [27–30]. Importantly, studies in social psychology have applied machine learning alongside SNA in an independent and complementary fashion [31, 32], yet few have utilized these methods in direct analytical combination—where attributes of a network are used as input predictors for a machine learning model. With a few exceptions, such direct use of these methods has been largely limited to the domain of cognitive neuroscience [33–35]. One unique study within psychology utilized data from 200 Twitter accounts to first construct a social network based on users' followers, extract features related to connectivity and engagement, and then apply these features to a support vector machine classifier in the prediction of self-report anxiety [36]. The results indicated that a model trained on social network features were predictive (AUC = 0.84) of anxiety disorder status [36]. Two additional studies focused on the predictive utility of egocentric network structural features using passively collected smartphone interaction data from $N = 53$ participants [37] and $N = 130$ participants [38] across 8 weeks to predict classification of low/high 5F-defined personality states. Egocentric networks focus on an individual (ego) and their direct connections with others (alters) to emphasize social standing from the perspective of the individual. Through comparison of egocentric networks across individuals within the broader social network, potentially informative structural differences can emerge. Broadly, the results of [37, 38] indicated strong and uniquely elucidative contributions of various egocentric network structural features to the prediction of personality. Bolstered by these studies and the documented, individual strengths of both machine learning and SNA, the present work sought to further apply the promise of this methodological marriage within the personality domain.

As a resource toward this goal, the current study extended the novel work of Gundogdu *et al.* (2017b), who implemented a social network analytic approach to interrogate the dynamics of personality [39]. In their study, $N = 54$ participants from an Italian research center wore a sociometric badge to record daily social interactions (counts of face-to-face contacts) over six weeks. Importantly, this collection method was continuous and passive in nature, providing uninterrupted monitoring of social behavior throughout the work day and eliminating the reliance on participants to actively log their interactions. The resulting data were used to create person-specific dyad, triad, and tetrad induced subgraph representations of interactions across different time intervals. To characterize the nodes in these subgraphs, participants were asked to complete EMA personality prompts three times a day which asked participants to reflect on recent interactions/behaviors with their coworkers. Each EMA item corresponded to a specific 5F personality trait. All responses ranged from "strongly disagree" to "strongly agree" on a 7-point Likert scale [39]. Logistic linear mixed models were used to predict personality state

transitions of an individual (e.g., "high" to "low" extraversion) as a function of the personality states of those with whom an individual interacted over a period of time. The results indicated that within-person variability in 5F personality traits was associated with variation in daily face-to-face interactions, with associations differing across traits [39]. For instance, participants were more likely to transition from a state of "low" agreeableness to a state of "high" agreeableness after interacting with two individuals with "high" agreeableness, whereas participants were more likely to transition to "high" openness when interacting with individuals with "low" openness [39].

## 1.1 Motivation

The work of Gundogdu *et al.* (2017b) is a valuable and creative first step in utilizing SNA to informatively blend trait-based measures of personality with the dynamics of situational context. To expand upon these efforts, the current exploratory work aimed to re-analyze their published dataset with the following key changes: (i) operationalize personality state change on a continuous scale instead of binary to more closely align outcome quantification with how the 5F (and WTT) literature conceptualizes personality manifestation (ii) focus more holistically on the utility of social network structural features to predict personality change instead of on the phenomenological insights gleaned from isolated graphlets of interactions, and (iii) embed SNA within a machine learning paradigm instead of employing a more traditional statistical modeling approach. In regard to this last point, few works [37, 38] have operationalized SNA in conjunction with machine learning within the personality literature. Under this relatively novel paradigm, the present study thus aimed to explore theoretical and practical extensions of these works and analytically complement the groundwork laid by Gundogdu *et al.* (2017b). Their work thus served both as a practical basis for model implementation and as a valuable opportunity to further our understanding of personality dynamics with ecologically valid data. To this end, the current work sought to incorporate a broader suite of SNA-derived features as well as focus on the ability to predict change in personality state rather than predicting the states themselves.

For the purposes of this study, utilizing SNA in direct interface with machine learning can provide insight into the relative importance of a wide array of social network structural features in the prediction of 5F personality state trajectories. As structural attributes of social networks capture different aspects of social processes, those attributes that are found to be most influential in a predictive model of personality state change may reflect prominent sociological contexts underlying or driving that change, ultimately highlighting foci for future hypothesis-driven research. Importantly, the analytic framework put forth in this study is intended to be used for hypothesis generation, with the goal of providing a means to investigate WTT-defined personality against the backdrop of evolving social environments. To this end, this endeavor was guided by the following aims:

i.  Using network representations of week-long social interactions alongside machine learning, quantify and summarize the idiographic utility of SNA-derived structural features for the prediction of 5F personality-specific change trajectories.

ii.  Explore and summarize the relative importance of SNA-derived structural features within and across the 5F personality traits.

iii.  Draw insights from resulting personality-specific profiles of relative SNA-derived structural feature importance to provide specific network attributes for further consideration and research.

iv.  Present a transparent, repeatable, and accessible quantitative framework for future application and refinement within the personality research domain.

## 2. Methods

### 2.1 Overview of study population and dataset

The current work utilized a previously published and publicly available dataset deposited online through Dryad [40]. The previous study was interested in utilizing wearable infrared passive sensing devices, specifically sociometric badges [41], in conjunction with structured EMA self-report, to interrogate the association between social interaction and personality state [39]. Accordingly, the dataset consisted of two separate but related timestamped logs across six weeks (January 30, 2012–March 9, 2012) for *N* = 54 Italian office employees (87% male; 84% Italian nationality). Participants varied in age from 23 to 53 years old (mean = 36.88, s.d. = 8.54). The first log listed device-detected instances of one-on-one, reciprocal employee interactions throughout the normal hours of the work week (Monday through Friday only). The second log regarded individual self-report EMA responses to reflections on recent (within the past half hour) personality-related behaviors. EMA prompts were given three times a day at 11:00 AM, 2:00 PM, and 5:00 PM, with domains and questions modeled after the Big Five Marker Scale (BFMS) [42] and the Ten Item Personality Inventory (TIPI) [43], and with responses ranging on a 7-point Likert scale from "strongly disagree" (1) to "strongly agree" (7). Under this framework, each EMA entry was associated with five scores, one for each of the Big Five personality states of extraversion, agreeableness, conscientiousness, emotional stability, and openness to experience. Moreover, each of these reported scores was an average of two TIPI item-level response scores that represented the respective personality state. Ultimately, this resulted in a dataset with 3,220 unique EMA responses quantified to represent an ecologically valid self-report summary of an individual's personality state and further contextualized with 248,749 contemporaneously recorded social interactions.

### 2.2 Data preprocessing and outcome operationalization

Using the provided EMA response logs, the data was first split based on the five-day work week. This resulted in six separate weekly logs which spanned the entirety of the data collection period. The EMA data in each week-based log was then processed independently to arrive at participant-specific operationalizations of personality state change for each respective week. Quantifying the average of absolute moment-to-moment changes in personality, the root mean square of successive differences (RMSSD) [44] was calculated for responses to extraversion, agreeableness, conscientiousness, emotional stability, and openness to experience. RMSSD is calculated as:

$$\sqrt{\frac{\sum_{i=1}^{N-1}\left(x_i - x_{i+1}\right)^2}{N-1}} \tag{1}$$

As illustrated in Formula (1), *i* is the measurement occasion, *N* is the total number of measurements, and *x* is the measurement value. This measure has been frequently used in the psychological literature to summarize dynamic change in affect [45]. Per-week, personality-specific RMSSD values for each participant thus represented the five independent modeling outcomes of interest. Table 1 summarizes the start and end dates, as well as the median, minimum, and maximum RMSSD values for each response across participants for each weekly log. This step is summarized in panel 1A of Fig 1.

### 2.3 Social network construction and visualization

Similar to the EMA data in 2.2, the available infrared passive sensing logs of participant interactions were split based on the five-day work week. Following this, the *networkX* (v2.4)

**Table 1. Summary of weekly EMA data by personality item.**

| Week | Start Date | End Date | RMSSD EXTRA | RMSSD AGREE | RMSSD CONSC | RMSSD STABL | RMSSD OPEN |
|------|-----------|----------|-------------|-------------|-------------|-------------|------------|
| | | | Median [Minimum, Maximum] | | | | |
| 1 | 30-Jan | 03-Feb | 1.24 [0.33, 3.15] | 0.85 [0.13, 1.87] | 0.84 [0.24, 2.02] | 0.76 [0.22, 2.04] | 0.91 [0.22, 2.67] |
| 2 | 06-Feb | 10-Feb | 1.11 [0.00, 2.61] | 0.85 [0.00, 2.33] | 0.73 [0.00, 2.12] | 0.79 [0.00, 2.48] | 0.97 [0.00, 2.39] |
| 3 | 13-Feb | 17-Feb | 1.21 [0.00, 3.47] | 0.75 [0.00, 1.99] | 0.71 [0.00, 2.22] | 0.80 [0.00, 1.78] | 0.93 [0.00, 3.11] |
| 4 | 20-Feb | 24-Feb | 1.18 [0.20, 2.66] | 0.83 [0.00, 2.37] | 0.68 [0.00, 1.97] | 0.73 [0.00, 2.50] | 0.84 [0.00, 3.48] |
| 5 | 27-Feb | 02-Mar | 1.04 [0.00, 3.30] | 0.80 [0.00, 2.02] | 0.75 [0.00, 2.86] | 0.66 [0.00, 2.19] | 0.93 [0.00, 2.14] |
| 6 | 05-Mar | 09-Mar | 1.18 [0.00, 2.42] | 0.80 [0.00, 1.89] | 0.65 [0.00, 2.05] | 0.78 [0.00, 1.75] | 0.93 [0.00, 2.30] |

*Note*. EMA data was split by the five-day work week (Monday-Friday) into six separate logs. The RMSSD of responses to each personality state item across the week was calculated for each participant. Values in the table reflect summarized RMSSD for the entire cohort ($N$ = 54) within each respective week. RMSSD = root mean square of successive differences; EXTRA = extraversion; AGREE = agreeableness; CONSC = conscientiousness; STABL = stability; OPEN = openness to experience.

package [46] in the Python programming language (v3.8.3) was leveraged. The *networkX* package allows for the in-depth study of networks by providing a broad toolkit to create, manipulate, and probe relationship structure, dynamics, and function. Specifically, this study used *networkX* to build and visualize six undirected, weighted graphs representing the network of social interactions of all participants within a given work week. A node in these graphs represented a participant, while each edge represented a logged reciprocated interaction between two participants. Edge weights between nodes were equivalent to the total count of interactions between participants within the designated week. It is important to note that the authors of the original data reported issues of detector reciprocity in a subset of the logged interactions—one detector in a pair would log a specific interaction while the other would not. To address this issue, the current study chose to model as edge weights the minimum possible number of interactions between every participant as determined by taking the lower summed count of interactions logged by the associated detectors in question. This ensured consistency in handling discrepancies between detectors. The resulting networks were visualized using the Fruchterman-Reingold algorithm [47]. This step is summarized in panel 1B-1 of Fig 1.

## 2.4 Network feature extraction

To holistically operationalize the context of social interaction, 26 structural and nodal features from the week-based social networks (see 2.3) were quantified using the *networkX* package. Table 2 provides an exhaustive list of each feature along with its associated scope, operational definition, and general contextual meaning. In an effort to maximize practical utility, interpretability, and accessibility, the theoretical intricacies of some of the more complex features exceed the scope of the current work. However, interested readers are highly encouraged to consult an excellent online textbook on the topic of social networks from which many of this study's contextual network interpretations were derived [48]. In summary, features were selected to capture unique aspects of (i) the overall social network structure of interactions, (ii) the individual's (node's) positioning within the broader network of interactions, and (iii) the local structural properties of the network from the egocentric perspective of the individual. Moreover, these features can be interpreted relative to the environmental context of this study's data—the workplace. As one example, reach efficiency quantifies the unique social value of an interaction in the network—in other words, the degree to which one employee interacts with another employee with connections that the other does not have. Thus, an employee with high reach efficiency could be thought to interact with a co-worker who interacts with several other co-workers that are more similar to each other (e.g., same project team,

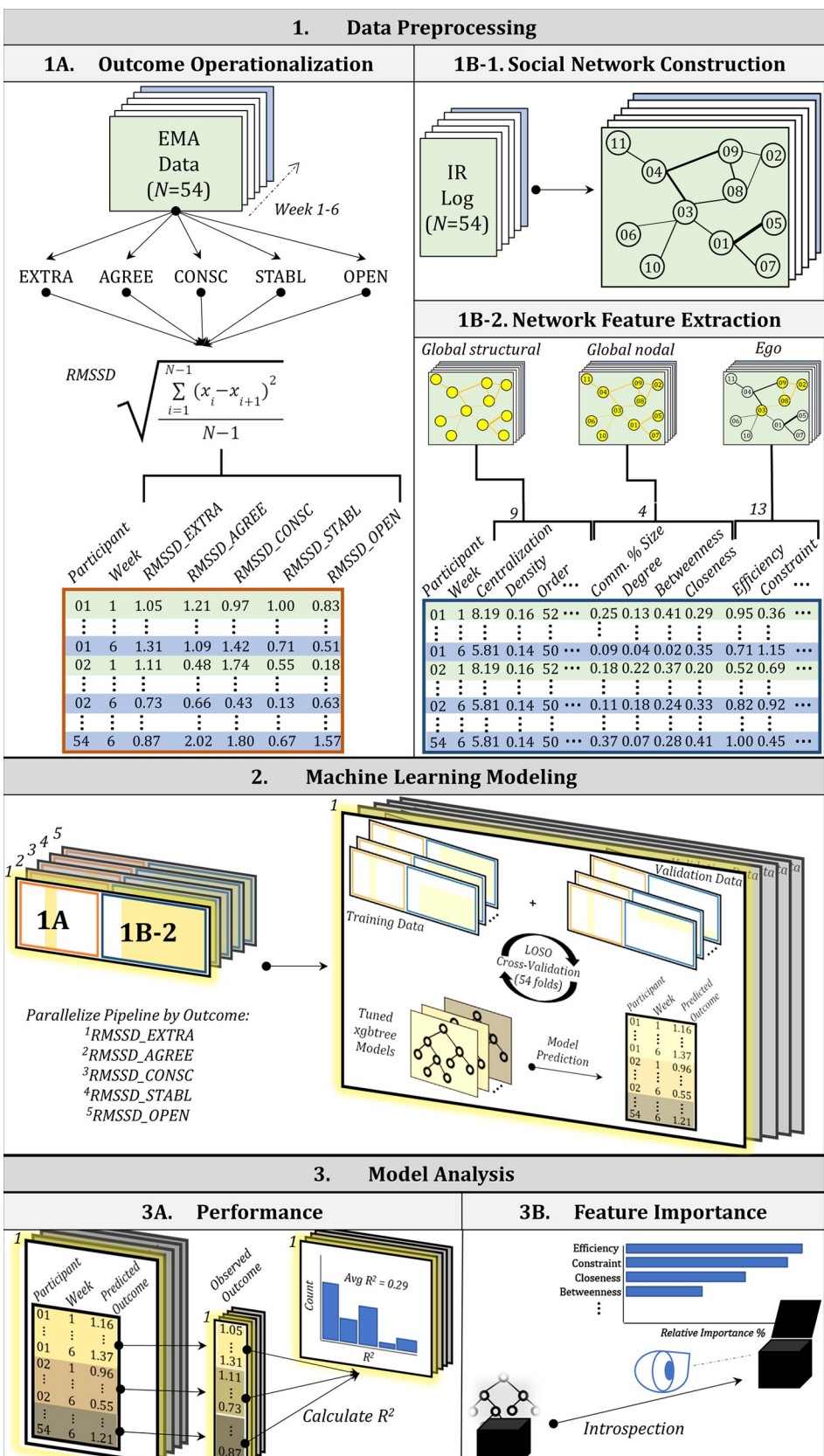

**Fig 1. Overview of analysis pipeline.** (1A) Derivation of outcome data for modeling. Raw EMA data is separated into six weeks and the RMSSD for each personality state self-report response within each week is calculated for each participant. (1B-1) Weekly cross-sectional social networks are constructed from the raw infrared passive sensing device log data. (1B-2) Features from the constructed networks are calculated (9 global structural, 3 global nodal, and 13 egocentric network features) to serve as predictors for the machine learning models. (2) The machine learning modeling framework is parallelized to independently predict each of the five personality states' RMSSD values. The model is trained on $N$-1 participants' network feature and outcome data and validated on a held-out participant's data. This is repeated 54 times such that each participant is held-out and trained using all other participants' data (LOSO cross-validation). A uniquely tuned model is validated for each fold and each model's predictions are saved. (3A) Individual model (fold) performance is assessed using variance explained ($R^2$), and average $R^2$ is calculated to assess overall performance of the machine learning framework across participants for each personality state outcome. (3B) Introspection of the models is performed via quantification of feature importance.

workspace proximity, personal interests), but for one reason or another do not interact directly with the employee. In this manner, the employee "extends" his or her work network (and the ability to transmit information) via a valuable interaction with a single co-worker. In total, this network approach thus sought to probe an array of social processes and phenomena and ultimately relate their presence and magnitude to individual, self-report conceptualizations of personality state.

To fully accomplish this goal, network features were derived from each complete (global) weekly network and from the construction of egocentric (ego) networks—subnetworks that include a focal node (the ego) and all of the nodes to whom the ego has a connection (one step removed)–for each individual across the six work weeks of the data collection period. Where the global network structural features are, by definition, consistent across individuals within a week, yet uniformly different across the cohort from one week to the next, global nodal and ego structural features differ among individuals both within and across weeks. The final prediction space for subsequent modeling (see 2.5), thus consisted of these 26 features across six weeks for $N$ = 54 individuals and resulted in 307 unique data points with which to predict associated within-week RMSSD values that represented dynamics of personality state change. This step is summarized in panel 1B-2 of Fig 1.

Please see S1 File for the entirety of this derived dataset along with feature-specific distributional statistics.

## 2.5 Machine learning modeling and analysis

A machine learning approach was employed to model a suite of extracted network features and explore the relative predictive merit of each within the context of personality state change. In this application, the dynamics of the model's learning process serve as a means with which to highlight potentially relevant aspects of social networks (and their associated phenomenology). Features found to be most important in informing the decision process of a well-performing model thus present as signals that can inform and direct further research efforts and hypotheses.

To achieve this, all predictive modeling and analyses were conducted using the R programming language (v4.0.2). Five separate and parallel eXtreme Gradient Boosting tree (xgbtree) models [50] were constructed, validated, and assessed with the *caret* library [51] to predict weekly person-specific RMSSD of (i) extraversion, (ii) agreeableness, (iii) conscientiousness, (iv) emotional stability, and (v) openness to new experience personality state self-report scores as a function of contextual social network structural features (see 2.4). Briefly, the xgbtree model operates by constructing decision trees in a sequential manner, where each subsequent tree in the sequence learns from the mistakes of its predecessor and updates the residual errors accordingly. This process, known as "boosting", converts what would normally be a set of

**Table 2. Extracted network features: Scope, definitions, and contextual meanings.**

| Feature | Scope | Operational Definition | Contextual Meaning |
|---|---|---|---|
| **Average Dyadic Redundancy–Ego** | Egocentric Network Structure | average across alters of how many of the other alters in the network are also tied to the alter of interest | *positional advantage/disadvantage*; to what degree does the ego engage in novel interactions to their benefit/detriment? |
| **Average Similarity–Ego** | Egocentric Network Structure | average number of neighbors shared by the ego and each of the alters | *shared interaction tendency*; to what degree do the alters directly interact with the same individuals as the ego? |
| **Betweenness Centrality** | Global Network Node | percentage of shortest paths, each connecting a unique node pair, where the target node is included | bridging position indicative of *control/power*; how frequently does the target node act as a liaison between individuals? |
| **Betweenness Centrality–Ego** | Egocentric Network Structure | percentage of shortest paths, each connecting a unique alter-alter pair, that pass through the ego | *social capital*; to what degree does the ego facilitate exchange and form interpersonal ties within their local neighborhood of interactions? |
| **Centralization (Degree)** | Global Network Structure | sum of the difference between the maximum node degree and the degree of all other nodes, divided by the order minus 1 | extent to which density/cohesion is organized around particular nodes; how *integrated* is the work community network overall? |
| **Centralization (Degree)–Ego** | Egocentric Network Structure | sum of the difference between the maximum node degree and the degree of all other nodes, divided by the order minus 1 | extent to which density/cohesion is organized around the ego and its alters; how *integrated* is the ego network overall? |
| **Closeness Centrality** | Global Network Node | reciprocal of the sum of the length (1/weight) of the shortest paths between the target node and all other nodes (normalized) | target node's *ability to easily spread and receive information*; how important/close is the individual to all other individuals within the network? |
| **Community Percent Size** | Global Network Node | From the Louvain modularity algorithm [49], the number of nodes that belong to the assigned cluster of the target node | *decoupling potential* of the target node; how many individuals comprise the subcommunity to which the target node belongs? |
| **Constraint–Ego** | Egocentric Network Structure | number of ego's connections that are to alters who are connected to one another (normalized) | *freedom* of ego's action as a function of the relationships among the alters; how independent is the ego within their local neighborhood of interactions? |
| **Degree Centrality** | Global Network Node | total number of edges connected to a target node (normalized) | overall magnitude of *integration* of the target node; how central/important is the individual within the work network? |
| **Density** | Global Network Structure | total number of observed ties divided by the total number of possible ties | global *cohesion*; how much constraint and opportunity exists for broader social interaction within the global network? |
| **Density–Ego** | Egocentric Network Structure | total number of observed ties divided by the total number of possible ties | local *cohesion*; how much constraint and opportunity exists for interaction among the ego and alters of the localized network? |
| **Diameter** | Global Network Structure | longest path length among all shortest paths from each node to all other nodes (weight agnostic) | network *spread*; how extensive is the network of work interactions? |
| **Effective Size–Ego** | Egocentric Network Structure | total number of alters less the average number of ties each alter has to other alters | *nonredundancy* of the ego; how impactful is the connectivity of the ego's local network? |
| **Efficiency–Ego** | Egocentric Network Structure | Effective size divided by the order | *impact* of the ego per unit investment in interaction; how much social gain results per tie? |
| **Geodesic Distance** | Global Network Structure | mean of the shortest path lengths (inverse weight) among all connected pairs | overall *connectedness* of the work network |
| **Geodesic Distance–Ego** | Egocentric Network Structure | mean of the shortest path lengths (inverse weight) among all connected pairs | overall *connectedness* of the ego's local network |
| **Order (N)** | Global Network Structure | number of nodes | *total individuals* interacting within the work community |
| **Number of Subcommunities** | Global Network Structure | total number of clusters found via the Louvain modularity algorithm [49] | *compartmentalization*; how divided are individuals within the work community? |
| **Number of Ordered Pairs** | Global Network Structure | total number of possible ties among all nodes | *opportunity* within the global network; how many potential interactions are there within the work community? |

*(Continued)*

**Table 2.** (Continued)

| Feature | Scope | Operational Definition | Contextual Meaning |
|---|---|---|---|
| **Number of Ordered Pairs–Ego** | Egocentric Network Structure | total number of possible ties among all nodes | *opportunity* within the egocentric network; how many potential interactions are there within the ego's neighborhood? |
| **Reach Efficiency–Ego** | Egocentric Network Structure | percentage of nodes that are within two directed steps of the ego divided by the order | *non-redundancy* of the ego; to what extent do the alters of the ego have unique connections that are not shared by the ego? |
| **Size (Number of Undirected Ties)** | Global Network Structure | total number of observed ties among all nodes | *social realization* within the global network; how many total interactions are there within the work community? |
| **Size (Number of Undirected Ties)–Ego** | Egocentric Network Structure | total number of observed ties among all nodes | *social realization* within the egocentric network; how many total interactions are there within the ego's neighborhood? |
| **Transitivity** | Global Network Structure | probability for the network to have adjacent nodes interconnected; observed number of closed triplets divided by the maximum possible number of closed triplets | *cliquiness* of the global network; is the work environment strongly connected? |
| **Transitivity–Ego** | Egocentric Network Structure | probability for the ego's neighborhood to have adjacent nodes interconnected; observed number of closed triplets divided by the maximum possible number of closed triplets | *cliquiness* of the ego; is the ego's local network strongly connected? |

*Note.* Each of the listed 26 network features were used as predictors in five separate machine learning modeling pipelines (one for each 5F state change outcome). Selected network features vary in scope, including features that quantify the overall structure of the summative weekly workplace social network (9 features), features that capture the position of a node/individual within the context of the summative weekly workplace social network (4 features), features that define the structure and position of each individual within their local (ego) network of interactions (13 features).

weak learners into a single strong learner. For context, the model representation and inference of xgbtree is identical to that of other tree-based learners such as the popular Random Forest model [52]; however, the underlying algorithm is distinct.

Each model was trained with leave-one-subject-out (LOSO) cross-validation. Under this scheme, all rows of data (1–6 rows; see 2.4) corresponding to a target individual were held-out while the remaining data across $N$-1 individuals (one fold) were used to train, hyperparameter tune, and test the model on the held-out individual's data. The following seven model hyper-parameters were tuned using the default grid search algorithm in *caret*: (i) the number of boosting iterations to perform (nrounds), (ii) the percent of training data to subsample for a given boosting iteration (subsample), (iii) the number of features to randomly subsample for each tree (colsample_bytree), (iv) the maximum depth allowed for each tree (max_depth), (v) the minimum weight required for each leaf node (min_child_weight), (vi) the minimum loss reduction required to further partition a leaf node (gamma), and (vii) the learning rate (eta). This was repeated $N = 54$ times to assess the model's performance both specifically within a fold (on a per individual basis) and holistically across all folds. This step is summarized in panel 2 of Fig 1.

Each of the five LOSO cross-validated models were assessed using $R^2$ (variance explained) and root mean square error (RMSE) at two levels of organization. The first level considers overall average performance of the model across all LOSO folds, while the second level considers performance of the model in predicting each individual's outcome as a function of all other individuals' network structural features. Performance results are summarized and presented in tabular, histogram, and linear graphical format (S1 File). This step is summarized in panel 3A of Fig 1. At the request of a reviewer, the authors additionally compared the overall average performance of each xgbtree model to two more simplistic and algorithmically distinct models: (i) a regularized generalized linear model and (ii) a k-nearest neighbors, clustering-based model using the same cross-validation approach and with default parameters in *caret*.

To introspect the resulting performance of the five models, the scaled feature importance in each model was calculated using the *varImp* function in *caret*. Intuitively, *varImp* operates by calculating differences in model error as a consequence of variable/feature permutation. Decreases in error represent improvements to the model and thus contribute to the overall magnitude of a feature's importance. This importance can then be scaled in relation to the importance of all other features for direct comparison purposes. This study reported both the scaled feature importance for each of the five personality-specific outcome models as well as the average overall feature importance across models. All features were ranked in order of descending average overall importance. This step is summarized in panel 3B of Fig 1.

The data preprocessing and network building Python script, as well as the R script for machine learning modeling and analysis, are available in S2 and S3 Files, respectively.

## 3. Results

### 3.1 Social networks

The graphs for each week-based social network are presented in Fig 2. As reported, the global network statistics of density, transitivity, and (especially) centralization change from week to week, thus indicating that the broader summative social context of the participants' work environment was not consistent across time. Against this shifting backdrop, individuals were not all interacting with the same co-workers or groups of co-workers, nor were they engaging in social interactions with the same frequency over time. The weekly networks express qualitatively appreciable variation in social engagement through time at both the individual and community level.

### 3.2 Model performance

Across individuals, Fig 3 illustrates that the predictive models built solely on social network structural features explained, on average, 28% of the variance in extraversion (A), 28% of the variance in agreeableness (B), 29% of the variance in conscientiousness (C), 30% of the variance in stability (D), and 29% of the variance in openness to experience (E), indicating that there were, on average, large predictive associations for each of the 5F constructs [53]. While promising, it is important to note that there was a large degree of interspecific variability in each of the personality-specific models. Moreover, Table 3 illustrates the high intraspecific variability (both in terms of variance explained and error) across personality models. For the majority of participants (37/55), models for some personalities performed very poorly while others performed very well (e.g., participant 509 with minimum $R^2$ = 0.04 and maximum $R^2$ = 0.72). Idiographically, the models were consistently informative across personality outcomes for approximately one-third (18/55) of the participants (see gray cells in Table 3). Models were defined as consistently informative if the worst performing personality model explained at least 5% of the variance ($R^2_{Min} \geq 0.05$) and the average normalized RMSE across models did not exceed 25% ($RMSE_{Avg} \leq 0.25$) of the model outcome's range of observed values (i.e, RMSSD ranging from 0.00 to 3.48; see Table 1). Under this operationalization, performance was consistently poor only in one instance (participant 538 with minimum $R^2$ = 0.00 and maximum $R^2$ = 0.04). Furthermore, in consideration of model error independently of $R^2$, predictions exhibited relatively small deviances ($RMSE \leq 0.25$) from the actual outcome values for 52/54 participants. These results holistically suggest the predictive capability of social network structural features in modeling individual personality state change.

For a comprehensive account of all in-fold (idiographic) predictions for each of the 5F personality state models, S4 File provides performance plots of observed versus predicted RMSSD values along with respective $R^2$ calculations.

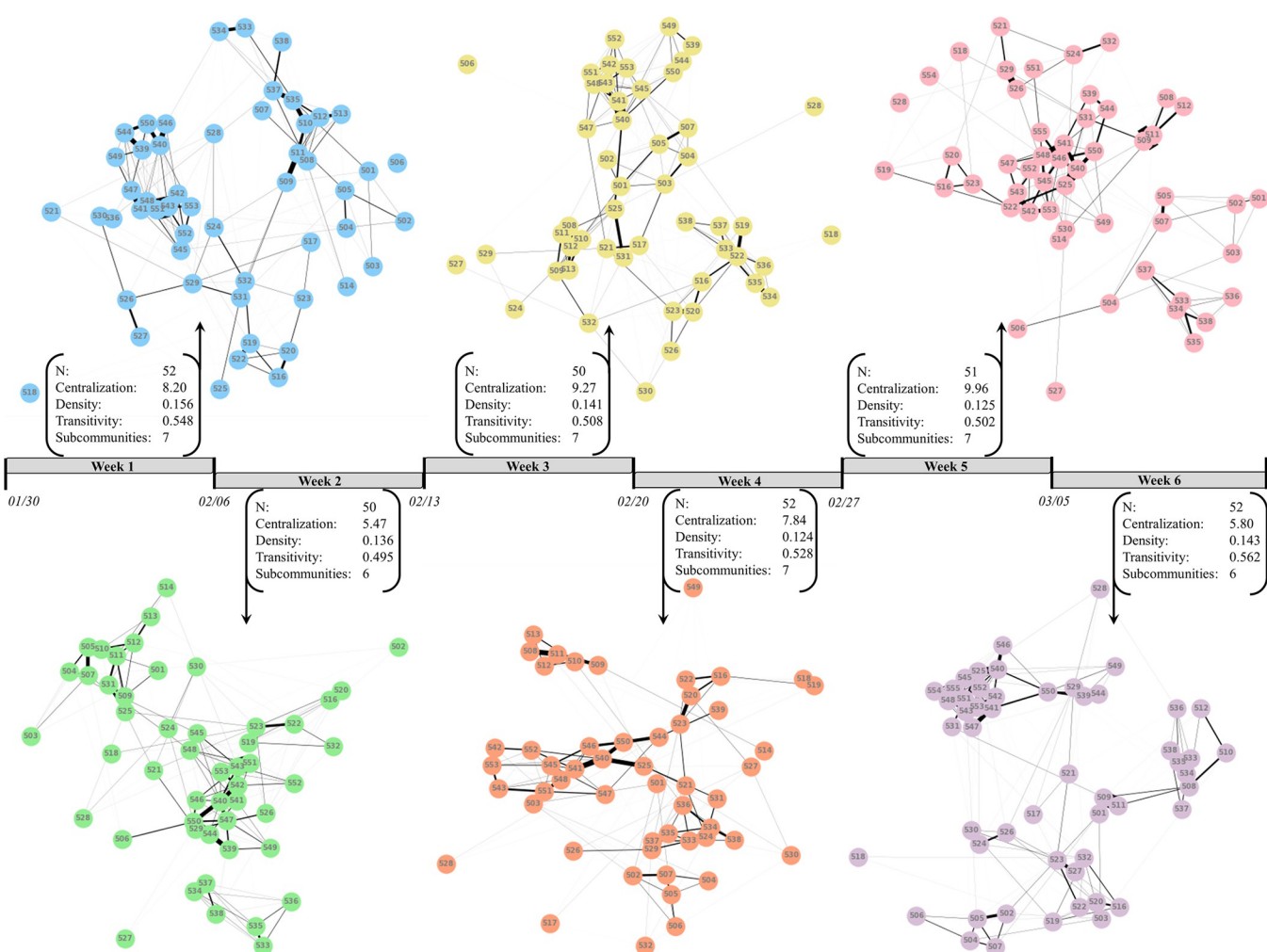

**Fig 2. Weekly undirected weighted networks of interactions.** Each of the six social networks represent summed infrared-detected interactions (undirected weighted edges) with individuals (nodes) across all work hours (Monday-Friday) for a given week. Nodes are labeled by participant ID and are consistent across networks. Edge thickness scales positively with the number of interactions. Summative network statistics of total number of nodes (N), degree centralization, density, transitivity, and the number of subcommunities are provided for each network.

*Post hoc* comparison in overall average performance for each of the above 5F xgbtree models along with their respective generalized linear model and k-nearest neighbor model implementations indicated that the xgbtree models consistently explained a larger percentage of the variance in personality state RMSSD relative to the generalized linear models, while variance explained in xgbtree models was comparable or greater in relation to all corresponding k-nearest neighbors models. Despite generally superior $R^2$, overall average RMSE was consistently highest (however marginally in most cases) among the xgbtree models. S5 File provides a table which details comparative performance among the models.

### 3.3 Feature importance

**3.3.1 Extraversion models.** As shown in Table 4, efficiency (100.00) and constraint of the ego within the ego network (80.30) were among the most important features in predicting change in self-report extraversion over time. The size of the ego network (58.47) was considerably more important for extraversion state change predictions relative to all other personality

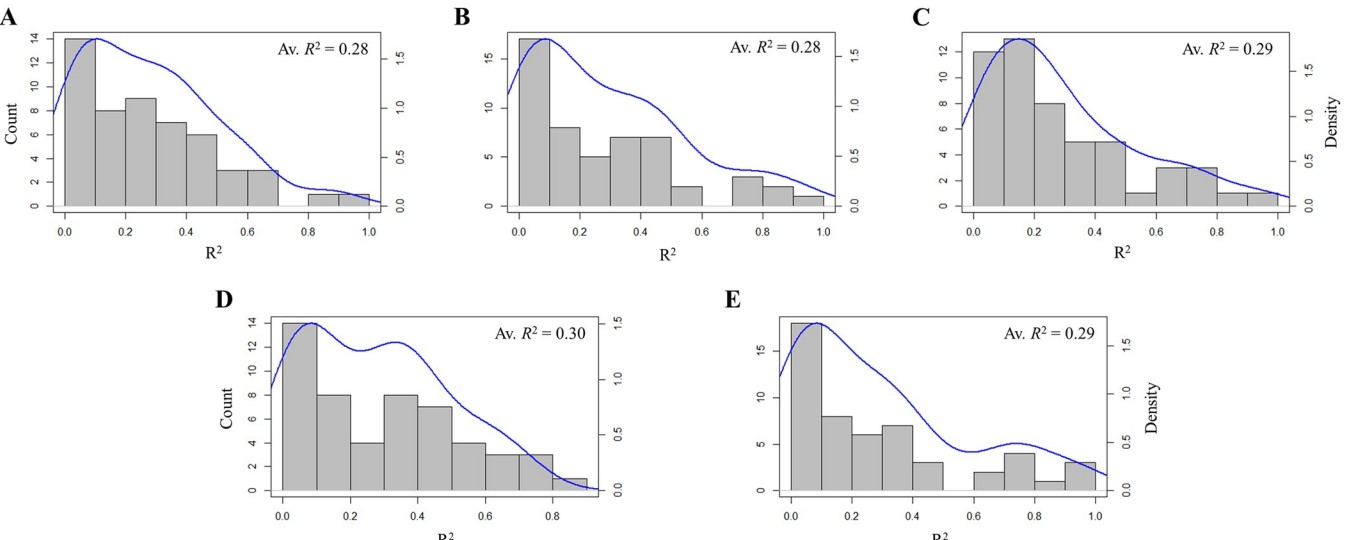

**Fig 3. Distribution of predictive performance across individuals for each five factor-based outcome model.** (A) extraversion models; (B) agreeableness models; (C) conscientiousness models; (D) stability models; (E) openness to experience models. Reported $R^2$ values represent the average $R^2$ across all LOSO fold performances ($N = 54$) for each respective model.

states (17.87 on average). No features were of much lower importance when compared to all other personality models.

**3.3.2 Agreeableness models.** As shown in Table 4, effective size of the ego network (100.00) and closeness centrality of the ego within the global workplace network (93.77) were the most important features predicting change in self-report agreeableness over time. Effective size was uniquely highly important relative to other personality states (45.94 on average), while closeness centrality was considerably more important for agreeableness state change predictions compared to all other personality state models (55.98 on average). Average similarity of the ego within the ego network (64.69) was also highlighted as more uniquely important relative to the other personality states (28.55 on average). However, no features were of much lower importance when compared to all other personality models.

**3.3.3 Conscientiousness models.** As shown in Table 4, efficiency (100.00), reach efficiency (95.65), and constraint (79.67) of the ego within the ego network were among the most important features in predicting change in self-report conscientiousness over time. Effective size of the ego network was considerably lower in importance (29.50) for conscientiousness models in relation to all other personality state models (63.56 on average). No features were of considerably higher importance when compared to all other personality models.

**3.3.4 Stability models.** As shown in Table 4, reach efficiency (100.00) and efficiency (85.03) of the ego within the ego network were among the most important features in predicting change in self-report stability over time. Density of the egocentric network was considerably lower in importance (18.71) for stability models in relation to all other personality state models (44.59 on average). No features stood out as being considerably higher in importance when compared to all other personality state models.

**3.3.5 Openness to experience models.** As shown in Table 4, efficiency of the ego within the ego network was the only structural network attribute among the most important features (importance > 75.00) predicting change in self-report openness to experience over time. Constraint (22.17) and betweenness centrality (22.25) of the ego within the ego network as well as closeness centrality of the ego within the global workplace network (33.48) were of uniquely

**Table 3. Overall, in-fold cross-validated performance across five factor predictive models.**

| | | $R^2$ | | | RMSE | | ID | | $R^2$ | | | RMSE | |
| ID | Min | Max | Avg. | Min | Max | Avg. | | Min | Max | Avg. | Min | Max | Avg. |
|---|---|---|---|---|---|---|---|---|---|---|---|---|---|
| 506 | 0.34 | 0.66 | 0.34 | 0.14 | 0.45 | 0.25 | 533 | 0.03 | 0.73 | 0.23 | 0.08 | 0.29 | 0.18 |
| 545 | 0.19 | 0.62 | 0.39 | 0.11 | 0.22 | 0.17 | 512 | 0.02 | 0.16 | 0.09 | 0.07 | 0.12 | 0.09 |
| 504 | 0.18 | 0.5 | 0.38 | 0.09 | 0.2 | 0.12 | 544 | 0.02 | 0.82 | 0.29 | 0.15 | 0.23 | 0.19 |
| 522 | 0.18 | 0.37 | 0.27 | 0.09 | 0.17 | 0.12 | 537 | 0.01 | 0.36 | 0.26 | 0.07 | 0.15 | 0.10 |
| 511 | 0.18 | 0.54 | 0.38 | 0.08 | 0.28 | 0.16 | 548 | 0.01 | 0.84 | 0.32 | 0.05 | 0.14 | 0.11 |
| 513 | 0.16 | 0.89 | 0.37 | 0.06 | 0.11 | 0.08 | 542 | 0.01 | 0.59 | 0.24 | 0.09 | 0.19 | 0.14 |
| 551 | 0.15 | 0.27 | 0.21 | 0.16 | 0.25 | 0.20 | 519 | 0.01 | 0.92 | 0.32 | 0.09 | 0.29 | 0.17 |
| 540 | 0.14 | 0.9 | 0.36 | 0.13 | 0.25 | 0.18 | 550 | 0.01 | 0.71 | 0.36 | 0.07 | 0.3 | 0.18 |
| 524 | 0.13 | 0.42 | 0.27 | 0.12 | 0.51 | 0.28 | 543 | 0 | 0.25 | 0.09 | 0.08 | 0.14 | 0.11 |
| 516 | 0.12 | 0.47 | 0.31 | 0.06 | 0.14 | 0.10 | 541 | 0 | 0.47 | 0.27 | 0.08 | 0.2 | 0.16 |
| 546 | 0.1 | 0.68 | 0.36 | 0.05 | 0.21 | 0.13 | 531 | 0 | 0.48 | 0.23 | 0.08 | 0.27 | 0.16 |
| 517 | 0.1 | 0.98 | 0.52 | 0.11 | 0.28 | 0.18 | 529 | 0 | 0.47 | 0.14 | 0.08 | 0.28 | 0.16 |
| 501 | 0.08 | 0.4 | 0.26 | 0.09 | 0.16 | 0.13 | 521 | 0 | 0.94 | 0.33 | 0.12 | 0.23 | 0.17 |
| 553 | 0.08 | 0.98 | 0.43 | 0.09 | 0.27 | 0.15 | 549 | 0 | 0.6 | 0.29 | 0.12 | 0.24 | 0.17 |
| 530 | 0.07 | 0.39 | 0.29 | 0.11 | 0.2 | 0.15 | 502 | 0 | 0.23 | 0.09 | 0.1 | 0.3 | 0.17 |
| 539 | 0.07 | 0.68 | 0.26 | 0.09 | 0.23 | 0.15 | 505 | 0 | 0.9 | 0.43 | 0.13 | 0.25 | 0.19 |
| 503 | 0.05 | 0.49 | 0.31 | 0.09 | 0.19 | 0.14 | 538 | 0 | 0.04 | 0.02 | 0.09 | 0.29 | 0.20 |
| 552 | 0.05 | 0.73 | 0.27 | 0.05 | 0.19 | 0.14 | 532 | 0 | 0.68 | 0.25 | 0.12 | 0.31 | 0.21 |
| 526 | 0.05 | 0.75 | 0.27 | 0.12 | 0.27 | 0.19 | 518 | 0 | 0.38 | 0.19 | 0.14 | 0.29 | 0.21 |
| 547 | 0.04 | 0.53 | 0.26 | 0.1 | 0.19 | 0.14 | 507 | 0 | 0.38 | 0.21 | 0.19 | 0.25 | 0.22 |
| 510 | 0.04 | 0.29 | 0.14 | 0.05 | 0.31 | 0.16 | 508 | 0 | 0.75 | 0.41 | 0.15 | 0.44 | 0.23 |
| 535 | 0.04 | 0.46 | 0.31 | 0.07 | 0.26 | 0.17 | 528 | 0 | 0.81 | 0.44 | 0.15 | 0.37 | 0.24 |
| 525 | 0.04 | 0.83 | 0.58 | 0.09 | 0.29 | 0.19 | 536 | 0 | 0.43 | 0.13 | 0.16 | 0.32 | 0.24 |
| 509 | 0.04 | 0.72 | 0.32 | 0.15 | 0.33 | 0.23 | 534 | 0 | 0.46 | 0.16 | 0.16 | 0.29 | 0.25 |
| 523 | 0.03 | 0.71 | 0.41 | 0.07 | 0.19 | 0.13 | 527 | 0 | 0.61 | 0.27 | 0.17 | 0.44 | 0.27 |
| 520 | 0.03 | 0.4 | 0.27 | 0.08 | 0.21 | 0.15 | 555 | – | – | – | 0.05 | 0.26 | 0.14 |
| 514 | 0.03 | 0.96 | 0.42 | 0.1 | 0.2 | 0.17 | 554 | – | – | – | 0.07 | 0.22 | 0.17 |

*Note.* Individuals are ranked highest to lowest based on minimum $R^2$ model performance ($R^2_{Min}$) and then by average normalized RMSE (RMSE$_{Avg}$) from lowest to highest. Highlighted cells represent individuals with whom each of the five models explained at least 5% of the variance ($R^2_{Min} \geq 0.05$) with an accompanying RMSE$_{Avg}$ across models that does not exceed 25% (RMSE$_{Avg} \leq 0.25$) of the model outcome's range of observed values (0.00–3.48).

low importance relative to other personality models (74.40, 51.49, and 71.05 on average, respectively). No features were of uniquely high importance relative to other personality models.

**3.3.6 Average across personality-specific models.** When considering all personality models together, efficiency (91.31), reach efficiency (75.02), and constraint (63.95) of the ego within the ego network, along with both closeness centrality of the ego in the global workplace network (63.54) and the effective size of the ego network (56.75) were the top five most important structural social network attributes influencing the prediction of personality state change over time.

## 4. Discussion

This study leveraged passively collected logs of workplace interaction alongside EMA data on personality-related behaviors from a previously published study on $N = 54$ office workers to characterize and compare the utility of social network structural features in the prediction of

**Table 4. Rank order of scaled feature importance scores across five factor predictive models.**

| Network Feature | EXTRA | AGREE | CONSC | STABL | OPEN | *Average* |
|---|---|---|---|---|---|---|
| Efficiency–Ego | 100.00 | 71.51 | 100.00 | 85.03 | 100.00 | **91.31** |
| Reach Efficiency–Ego | 56.04 | 60.21 | 95.65 | 100.00 | 63.19 | **75.02** |
| Constraint–Ego | 80.30 | 70.00 | 79.67 | 67.61 | 22.17 | **63.95** |
| Closeness Centrality | 71.78 | 93.77 | 64.25 | 54.41 | 33.48 | **63.54** |
| Effective Size–Ego | 47.72 | 100.00 | 29.50 | 62.61 | 43.91 | **56.75** |
| Betweenness Centrality–Ego | 54.69 | 60.23 | 46.94 | 44.11 | 22.25 | **45.65** |
| Betweenness Centrality | 57.14 | 25.50 | 47.60 | 61.80 | 33.87 | **45.18** |
| Community Percent Size | 37.78 | 23.65 | 59.92 | 50.67 | 46.35 | **43.67** |
| Transitivity–Ego | 36.44 | 32.43 | 48.48 | 48.55 | 33.35 | **39.85** |
| Density–Ego | 32.81 | 48.71 | 48.47 | 18.71 | 48.35 | **39.41** |
| Average Similarity–Ego | 26.16 | 64.69 | 32.35 | 42.60 | 13.08 | **35.78** |
| Size–Ego | 58.47 | 30.26 | 23.74 | 15.98 | 1.50 | **25.99** |
| Centralization–Ego | 24.76 | 40.75 | 26.60 | 21.33 | 11.66 | **25.02** |
| Average Dyadic Redundancy–Ego | 15.93 | 37.16 | 21.84 | 20.76 | 17.37 | **22.61** |
| Degree Centrality | 32.83 | 33.08 | 22.27 | 12.19 | 12.64 | **22.60** |
| Week | 11.94 | 20.34 | 10.93 | 17.09 | 0.24 | **12.11** |
| Size | 18.25 | 9.84 | 8.86 | 6.23 | 7.32 | **10.10** |
| Centralization | 7.41 | 9.39 | 9.32 | 6.83 | 8.52 | **8.29** |
| Geodesic Distance–Ego | 13.51 | 3.37 | 4.31 | 0.47 | 9.81 | **6.29** |
| Transitivity | 9.95 | 0.00 | 6.90 | 0.45 | 0.00 | **3.46** |
| Density | 5.92 | 3.61 | 4.07 | 2.01 | 0.00 | **3.12** |
| Geodesic Distance | 0.44 | 3.36 | 0.74 | 2.50 | 3.94 | **2.20** |
| Number Ordered Pairs–Ego | 6.62 | 3.66 | 0.00 | 0.00 | 0.00 | **2.06** |
| Diameter | 4.43 | 0.00 | 2.99 | 2.30 | 0.00 | **1.94** |
| Number of Nodes | 7.05 | 0.00 | 1.27 | 0.38 | 0.00 | **1.74** |
| Number of Ordered Pairs | 1.21 | 0.00 | 2.54 | 0.51 | 0.00 | **0.85** |
| Number of Subcommunities | 0.00 | 0.00 | 0.00 | 0.14 | 0.00 | **0.03** |

*Note.* Network features are listed in order of descending average importance across personality traits. All importance scores are relative and thus standardized with a scale of 0 to 100. Highlighted cells represent features that were among the most important (>75.00) for the respective personality state model. Underlined values indicate features which stand out as being notably higher or lower in importance for the given personality model relative to all other models. EXTRA = extraversion; AGREE = agreeableness; CONSC = conscientiousness; STABL = stability; OPEN = openness to experience.

personality state change within and among 5F personality constructs. Through a combination of both SNA and machine learning, this research aimed to present a relatively uncommon exploratory workflow and demonstrate its ability to operationalize and highlight potentially significant social processes within the context of the personality literature. Model performance (in-fold) was highly heterogeneous within and across individuals (Table 3); however, results at the cohort level (out-of-fold) reflect an average of above-chance predictive performance (Fig 3). From these models, efficiency—the proportion of non-redundant ties, signifying an individual's diversity of social interactions in an SNA framework—was found to be consistently important in predicting change across all 5F personality states, while several other features such as effective size, closeness centrality, and constraint exercised a uniquely high or low influence within specific personality state predictions. Most broadly, these findings bolstered past findings which found predictive merit of egocentric network structural features in personality modeling [37, 38]. Moreover, the results specifically highlighted efficiency, reach efficiency, and constraint within workplace egocentric networks, indicating that shifting social

contexts which particularly describe the diversity and shared interactions of people may be important indicators of personality fluctuation. Furthermore, this has implications for future research which may benefit from exploring these features to profile individual personality constructs and characterize how they change over time.

Each of the 5F models was capable of accounting for 28–30% of the variance in individual personality state change on average across the cohort (Fig 3). Given the complexity of the phenomena in question (i.e., personality change), the ability to obtain a predictive signal using only network structural measures was promising. While model performance at the level of the individual was highly variable (Table 2), the modeling framework only performed poorly across all personality state outcomes for one participant (ID 538). Additionally, in over 50% of the cohort, at least one dimension of personality state change was substantively ($R^2 > 0.5$) explained by social network structural features. The results simultaneously speak to the challenges of modeling constructs that are innately heterogeneous and suggest utility in employing SNA structural operationalizations alongside machine learning to model personality. Taken together, the cohort-level performance of the five machine learning models justified a closer inspection of which network features were being utilized to predict personality state change.

In this study, model introspection specifically concerned the relative predictive importance of each network structural feature. As each of these features capture different social situations or processes, those features found to be more important in predicting change in personality state would potentially suggest a phenomenological linkage between the operationalized social dynamic and the specific personality state modeled as an outcome. Ultimately, highlighting these social "determinants" through definition of network feature profiles may direct the efforts of future research endeavors. Feature importance results across the five models implicated (i) efficiency, (ii) reach efficiency, (iii) constraint, (iv) closeness centrality, and (v) effective size of the egocentric network as the most important network features in the prediction of personality state change on average (Table 4). First, the ranking broadly indicates that the majority of the most influential structural attributes are related to the egocentric network, rather than the global workplace network or the individual embedded within this global scope. The importance pattern suggests that understanding how individuals are forming direct ties in their local neighborhood of interactions offers more predictive value into personality state change than how the community is forming ties as a whole. Second, the specific prominence of efficiency and closeness centrality echoes previous work where measures of centrality and efficiency were found to outperform measures of transitivity [38]. In the current study, transitivity within the global and egocentric networks was comparatively low in importance. Third, and in particular consideration of the office setting, the importance of efficiency and closeness centrality—measures that quantify the influence/value of ties and the ability to spread and receive information, respectively—speaks to the potential preference or need to disseminate information quickly to well-connected people. For example, one research study found that individuals concerned with how they were perceived by others within the workplace (i.e. those who were highly conscientious self-monitors) tended to occupy more central positions within the workplace network [54].

Focusing more specifically on a few of the 5F personality state-specific results, the size of the egocentric network was of a noticeably higher relative importance (58.7; rank 4) in predicting extraversion state change compared with all other personality states (17.9; rank 15 on average). Previous work has specifically demonstrated a positive correlation between extraversion and egocentric network size [55]. Relatedly, the importance of closeness centrality (71.78; rank 3), while not the highest among the personality models, was still highly influential in predicting extraversion state change. Like egocentric network size, several measures of centrality, including closeness centrality, have previously been found to positively correlate with extraversion

[37, 56]. The importance of constraint—the extent to which the ego's connections are to others who are themselves connected to each other—signifies that the ego's interactions with others who either have a higher number of shared (high constraint) or a higher number of distinct (low constraint) interactions from the ego are predictive of extraversion. While this study does not assess directionality of the feature's importance on the model's prediction, the purported association of extraversion with the "diversity" of ties (similar to what is captured by efficiency) seems reasonable whether it is positive or negative. Furthermore, the intuitive hypothesis can be made that more highly extraverted behaviors may invite lesser constraint through willingness to interact with a broader array of individuals; however, further research would be needed to test whether this is the case.

In consideration of agreeableness, previous research has found that more agreeable people tend to form "small worlds" or short chains of interactions connecting individuals [38]. Structurally, this phenomenon is partly characterized by short distances (larger number of interactions) between nodes (individuals). In line with this, current analyses found that closeness centrality was particularly important (93.77; rank 2) in predicting agreeableness state change. Closeness centrality was considerably higher in importance when compared to all other personality states (55.98; rank 5 on average). Moreover, the connection between small world networks, closeness centrality, and agreeableness has been documented specifically within the workplace domain. One study examined small world networks among CEOs and their employees and found a positive, significant association between customer satisfaction and the interaction term for CEO agreeableness and closeness centrality [57]. The current work also found a uniquely higher relative importance for average similarity—the degree to which alters interact with the same individuals as the ego—within the egocentric network (64.69; rank 5). Compared with all other personality states (28.55; rank 11.25 on average), the importance of egocentric similarity for agreeableness speaks to the findings of a Facebook study where people, particularly males, with similarity in agreeableness, had stronger connectedness relative to others [58]. The unique implications for agreeableness similarity on the resulting structure of egocentric networks in males may partially explain the heightened predictive utility of this feature in the current study's cohort.

Turning lastly to openness, the majority of the centrality measures for both the global workplace and egocentric networks had relatively low importance in the openness models relative to all other personality constructs. This pattern makes some intuitive sense given that those who are open to new experiences may be more likely to interact with a wider array of people regardless of other socially relevant factors. Accordingly, the degree to which one is well-connected or central within the social network may carry less significance. However, the fact that efficiency solely dominates the prediction dynamics of this personality state could be a function of both the most consequential manifestation of openness behaviors within the workplace environment (e.g., openness to interact with individuals regardless of research group/department, thus influencing non-redundancy of ties) in combination with this potentially less informative signal in centrality. Importantly, an individual can be efficient without being central (in the strict sense of degree, betweenness, and closeness, but see [59]) thus this combination may be a unique signal for the openness personality state within the work setting. Differing from the current results, one study found that individuals with a higher openness to experience tend to more likely act as intermediaries between previously unconnected individuals [25]. This suggests that betweenness centrality should have some predictive merit; however, for both the global and egocentric networks, it was found to be of low importance (22.25; rank 9 and 33.87; rank 6) relative to all other personality constructs. Future research may benefit from focusing on the relative efficiency and centrality profile of openness (and personality more broadly) to see if the current results generalize to other cohorts both inside and outside the workplace setting.

This exploratory work has several important limitations. First, feature importance is a scalar measure, thus the current analysis was unable to probe the directionality of a feature's influence on a model's prediction. More complex model introspection techniques such as SHapely Additive exPlainers (SHAP) [60, 61] or LIME [62] may be employed in future efforts on larger datasets to more reliably ascertain potentially meaningful magnitudes of predictive influence. Relatedly, there is an inherent tradeoff between the complexity and interpretability of any "black box", machine learning-based approach. While the ability to peer inside these models (as mentioned in the first point above) has partially mitigated this tradeoff and has allowed researchers to contextualize and detail model performance within the purview of real-world phenomena, any interpretation outside the model's demonstration of predictive merit should be treated as hypothesis-generating and exploratory rather than hypothesis-testing and confirmatory. Indeed, the current exploratory work performed introspection on a model that, despite being easy to implement in practice, is algorithmically complex and thereby unable to provide the transparency of a more traditional statistical model.

Third, it is important to recognize that this approach highlights potentially fruitful patterns of network structure that inform personality state change; however, it cannot be used to ascertain causal relationships. Model importance does not necessarily relate to phenomenological importance; thus, results should be used as a guide to inform future research inquiries rather than as a direct translation of real-world phenomenology. Third, the cohort was largely homogenous, consisting mostly of Italian male researchers working in an office environment. Accordingly, the findings may not reflect processes that occur in the general population or across disparate contexts. Nevertheless, this work offers specialized insight into the network features that drive personality dynamics in an office setting. Fourth, a commonly cited limitation in passive sensing social network studies [37] is that the data did not account for interactions involving individuals who have not participated in the study. Lastly, behavioral reports during non-working hours were not collected and could therefore not be analyzed to provide a continuous account of personality state change from one day to the next.

Despite these limitations, the results are valuable in that they contribute additional proof of promise in the application of machine learning-based network modeling to personality—a literature which is currently still relatively small and underdeveloped. The exploratory modeling paradigm presented in this work is easily scalable to larger datasets and can be reproduced handily with only a few software packages in Python and R (see S2 and S3 Files). The current research has bolstered previous findings in support of the especially informative nature of egocentric social network features and has particularly implicated measures of efficiency, constraint, and closeness centrality as potentially fruitful markers of workplace personality dynamic change. Further exploration into specific structural network features of personality may yield unique insights that practically influence workplace organization and efficiency as well as theoretically inform broader facets of mental health and the human social condition.

## Supporting information

**S1 File. Network dataset and variable distributions.**
(XLSX)

**S2 File. Python script to build networks and extract features.**
(DOCX)

**S3 File. R script to perform machine learning modeling.**
(DOCX)

**S4 File. Plots of idiographic model predictions.**
(XLSX)

**S5 File. Comparative performance of disparate machine learning algorithms.**
(DOCX)

## Acknowledgments

We would like to thank Dr. Didem Gundogdu and colleagues at the Bruno Kessler Institute in Trento, Italy for the data utilized in this study. Their collection and subsequent publication of the data for public research use on Dryad made this work possible.

## Author Contributions

**Conceptualization:** Damien Lekkas.

**Formal analysis:** Damien Lekkas, Nicholas C. Jacobson.

**Methodology:** Damien Lekkas, Nicholas C. Jacobson.

**Project administration:** Damien Lekkas.

**Software:** Damien Lekkas.

**Supervision:** Nicholas C. Jacobson.

**Visualization:** Damien Lekkas.

**Writing – original draft:** Damien Lekkas, Joseph A. Gyorda.

**Writing – review & editing:** Damien Lekkas, Joseph A. Gyorda, Erika L. Moen, Nicholas C. Jacobson.

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
