## [Decision Letter · Decision Letter 0]

23 Aug 2022

PONE-D-22-21274Using passive sensor data to probe associations of social structure with changes in personality: A synthesis of network analysis and machine learningPLOS ONE

Dear Dr. Lekkas,

Thank you for submitting your manuscript to PLOS ONE. After careful consideration, we feel that it has merit but does not fully meet PLOS ONE’s publication criteria as it currently stands. Therefore, we invite you to submit a revised version of the manuscript that addresses the points raised during the review process.

ACADEMIC EDITOR: Two detailed reports are received on the manuscript and both the reviewers have raised critical points which needs to be addressed.

We look forward to receiving your revised manuscript.

Kind regards,

Dr. Rajesh Kumar

Academic Editor

PLOS ONE

Journal Requirements:

"This work was funded by an institutional grant from the National Institute on Drug Abuse (NIDA-5P30DA02992610)."

"DL, JAG, and NCJ received funding through a National Institute on Drug Abuse (NIDA; https://nida.nih.gov/) institutional grant (5P30DA02992610). The funders had no role in study design, data collection and analysis, decision to publish, or preparation of the manuscript."

Reviewers' comments:

Reviewer's Responses to Questions

**Comments to the Author**

1. Is the manuscript technically sound, and do the data support the conclusions?

Reviewer #1: Partly

Reviewer #2: Partly

2. Has the statistical analysis been performed appropriately and rigorously? 

Reviewer #1: No

Reviewer #2: No

3. Have the authors made all data underlying the findings in their manuscript fully available?

Reviewer #1: Yes

Reviewer #2: Yes

4. Is the manuscript presented in an intelligible fashion and written in standard English?

Reviewer #1: Yes

Reviewer #2: Yes

5. Review Comments to the Author

Reviewer #1: Manuscript: Using passive sensor data to probe associations of social structure with changes in personality: A synthesis of network analysis and machine learning

This work leverages data from a previously published study to combine passively collected wearable sensor information on face-to-face, workplace social interaction with ecological momentary assessment of personality state. Data from 54 individuals across six weeks was used to explore the relative importance of social network structural features in predicting changes in the Big Five (5F) personality states. Using only SNA-derived features from wearable sensor data, models explained, on average, approximately 28-30% of the variance in individual personality state change. Model introspection implicated egocentric features as the most influential predictors across 5F-specific models with network efficiency, constraint, and effective size measures among the most important. Feature importance profiles for each 5F model partially echoed previous empirical findings. Results support future efforts focusing on egocentric components of SNA and suggest particular investment in exploring efficiency measures to model personality fluctuations within the workplace setting.

My observations and comments are as follows:

1. The quality of figures/images requires substantial improvement. At present they are quite blurry and not visible clearly.

2. Include a motivation subsection after the introduction section and highlight the point(s) which you saw were missing from the discussed literature and motivated you to undertake this research.

3. Try to organize the paper by including relevant sections and subsections. At present there are number of sections but the content in them is very limited.

4. How the machine learning is applied to the present problem is not clear to the reviewer.

5. Discuss the AI technique in more detail (which you have used) and declare/highlight its parameter settings clearly.

6. Enrich the section "Data preprocessing and outcome operationalization" with proper mathematical equations. Further, define "networkX " properly before using/applying it. Check the font of sentences from 264-272 lines.

7. The simulation results are not sufficient and the authors are encouraged to deepen their simulation study.

8. The efficacy of the proposed approach can only be judged if its performance is compared with some other machine learning technique(s). Authors are encouraged to work on this aspect.

9. Discuss the computational complexity of the proposed method.

10. Include a flowchart or pseudo code which shows the sequences of the processing steps involved in your proposed method.

Reviewer #2: The manuscript needs editing to explain few points mentioned in the attached file. Abstract should be rewritten.

Authors need to go through the attached file to improve the manuscript. Conclusion and discussion section needs attention and reformulation is needed.

6. PLOS authors have the option to publish the peer review history of their article (what does this mean?). If published, this will include your full peer review and any attached files.

Reviewer #1: No

Reviewer #2: No

---

## [Author Response · Author response to Decision Letter 0]

5 Oct 2022

Dear Dr. Kumar, Academic Editor, PLoS ONE:

Thank you for the opportunity to resubmit our manuscript, “Using passive sensor data to probe associations of social structure with changes in personality: A synthesis of network analysis and machine learning”, for consideration in PLoS ONE. We are appreciative of the reviewer’s time and attention to our work and found their comments and suggestions to be very helpful in the process of addressing key deficiencies and polishing the content and presentation of our work. Our response to each of the reviewer’s points is enumerated below:

Reviewer #1: 

1. The quality of figures/images requires substantial improvement. At present they are quite blurry and not visible clearly.

We have increased the DPI for each of our figures and have increased the font size where possible to improve readability. We have also rearranged and reformatted aspects of Figure 2 for increased clarity.

2. Include a motivation subsection after the introduction section and highlight the point(s) which you saw were missing from the discussed literature and motivated you to undertake this research.

We appreciate the reviewer’s suggestion and the opportunity to reorganize our Introduction for clarity. We have reworked and restructured our Introduction which now includes a Motivation subsection. To specifically address the reviewer’s comment regarding the inclusion of what we deemed to be missing from the literature, we have also added the following to our new Motivation subsection:

“In regard to this last point, few works have operationalized SNA in conjunction with machine learning within the personality literature. Under this relatively novel paradigm, the present study thus aimed to explore theoretical and practical extensions of these works and analytically complement the groundwork laid by Gundogdu et al. (2017b). Their work thus served both as a practical basis for model implementation and as a valuable opportunity to further our understanding of personality dynamics with ecologically valid data. To this end, the current work sought to incorporate a broader suite of SNA-derived features as well as focus on the ability to predict change in personality state rather than predicting the states themselves.”

3. Try to organize the paper by including relevant sections and subsections. At present there are a number of sections but the content in them is very limited.

We thank the reviewer for their comment which has led us to reorganize our Methods section. Specifically, we have combined sections 2.5 - 2.8 into one section entitled “Machine learning modeling and analysis”.

4. How machine learning is applied to the present problem is not clear to the reviewer.

For clarity, we have added the following to section 2.5:

“A machine learning approach was employed to model a suite of extracted network features and explore the relative predictive merit of each within the context of personality state change. In this application, the dynamics of the model’s learning process serve as a means with which to highlight potentially relevant aspects of social networks (and their associated phenomenology). Features found to be most important in informing the decision process of a well-performing model thus present as signals that can inform and direct further research efforts and hypotheses.”

5. Discuss the AI technique in more detail (which you have used) and declare/highlight its parameter settings clearly.

We have added the following details on the machine learning algorithm as well as the hyperparameters used to section 2.5:

“Briefly, the xgbtree model operates by constructing decision trees in a sequential manner, where each subsequent tree in the sequence learns from the mistakes of its predecessor and updates the residual errors accordingly. This process, known as “boosting”, converts what would normally be a set of weak learners into a single strong learner. For context, the model representation and inference of xgbtree is identical to that of other tree-based learners such as the popular Random Forest model; however, the underlying algorithm is distinct.”

“The following seven model hyperparameters were tuned using the default grid search algorithm in caret: (i) the number of boosting iterations to perform (nrounds), (ii) the percent of training data to subsample for a given boosting iteration (subsample), (iii) the number of features to randomly subsample for each tree (colsample_bytree), (iv) the maximum depth allowed for each tree (max_depth), (v) the minimum weight required for each leaf node (min_child_weight), (vi) the minimum loss reduction required to further partition a leaf node (gamma), and (vii) the learning rate (eta).”

6. Enrich the section "Data preprocessing and outcome operationalization" with proper mathematical equations. Further, define "networkX " properly before using/applying it. Check the font of sentences from 264-272 lines.

To make our methods more explicit we have added the formula for RMSSD to section 2.2. We have also added the following sentence to the first paragraph of section 2.3 where we further define and contextualize the networkX Python package:

“The networkX package allows for the in-depth study of networks by providing a broad toolkit to create, manipulate, and probe relationship structure, dynamics, and function. Specifically, this study used networkX to build and visualize six undirected, weighted graphs representing the network of social interactions of all participants within a given work week.“

We appreciate the reviewer’s attention to detail regarding our formatting for our Table 1 footnote (Lines 264-271). We have updated the alignment and layout to be double spaced in adherence to the Journal’s formatting guidelines for including embedded Table footnotes in-text. We have also updated the footnotes for Tables 2 and 3 similarly.

7. The simulation results are not sufficient and the authors are encouraged to deepen their simulation study.

We would like to clarify that our study was not a simulation study as we built networks from empirically collected social interaction data. Nevertheless, we have added further information regarding the depth of our analysis to Methods. Please see our responses to comments 5 and 6 above for specific additions. We have also expanded the scope and context of our modeling through implementation of two additional model types. Please see our response to comment 8 below for specifics.

8. The efficacy of the proposed approach can only be judged if its performance is compared with some other machine learning technique(s). Authors are encouraged to work on this aspect.

We appreciate the reviewer’s comment. While the goal of this work was exploratory and not meant to derive an optimally performing model, we agree with the reviewer that it would be better to show how our chosen algorithm compares to other most commonly used model types. As such, we reran our analysis pipeline twice using (1) a generalized linear model with elastic net penalty (glmnet) and (2) a clustering-based algorithm (knn) with default parameters. These two models were selected for comparison because they are commonly implemented in the literature and parse the data in disparate ways; our chosen algorithm (xgbtree) uses a boosted decision tree-based approach while glmnet is a basic linear model and knn utilizes clustering. The average performance results across folds is reported in the following table:

As shown in the table (see response letter file), the chosen xgbtree algorithm consistently explains a larger proportion of the variance (R2) when compared with a more simple linear model (glmnet) and performs comparatively or better in relation to a clustering-based algorithm (knn). We have added this table as a supplementary file (S5_File.docx). In addition we have added the following to Methods:

“At the request of a reviewer, the authors additionally compared the overall average performance of each xgbtree model to two more simplistic and algorithmically distinct models: (i) a regularized generalized linear model and (ii) a k-nearest neighbors, clustering-based model using the same cross-validation approach and with default parameters in caret.“

And we have added the following to Results, section 3.2 Model Performance:

“Post hoc comparison in overall average performance for each of the above 5F xgbtree models along with their respective generalized linear model and k-nearest neighbor model implementations indicated that the xgbtree models consistently explained a larger percentage of the variance in personality state RMSSD relative to the generalized linear models, while variance explained in xgbtree models was comparable or greater in relation to all corresponding k-nearest neighbors models. Despite generally superior R2, overall average RMSE was consistently highest (however marginally in most cases) among the xgbtree models. Supporting File S5 provides a table which details comparative performance among the models.“

9. Discuss the computational complexity of the proposed method.

We appreciate the reviewer’s suggestion to discuss the complexity of our chosen machine learning approach. We believe that the complexity is best discussed as a limitation, thus we have included the following in our Discussion:

“Relatedly, there is an inherent tradeoff between the complexity and interpretability of any “black box”, machine learning-based approach. While the ability to peer inside these models (as mentioned in the first point above) has partially mitigated this tradeoff and has allowed researchers to contextualize and detail model performance within the purview of real-world phenomena, any interpretation outside the model’s demonstration of predictive merit should be treated as hypothesis-generating and exploratory rather than hypothesis-testing and confirmatory. Indeed, the current exploratory work performed introspection on a model that, despite being easy to implement in practice, is algorithmically complex and thereby unable to provide the transparency of a more traditional statistical model. “ 

10. Include a flowchart or pseudocode which shows the sequences of the processing steps involved in your proposed method.

We thank the reviewer for their excellent suggestion and appreciate the opportunity to succinctly summarize our pipeline. We have added the figure on the following page (see response letter file) to the manuscript which illustrates each major step of analysis. For transparency and reproducibility, we also provide the full code within the supplementary files.

  

Reviewer #2:

1. The manuscript needs editing to explain a few points mentioned in the attached file. Abstract should be rewritten. Authors need to go through the attached file to improve the manuscript. Conclusion and discussion section needs attention and reformulation is needed.

We greatly appreciate the reviewer’s time and consideration of our manuscript. We have addressed each of the points mentioned in the file below.

2. The abstract can be re-written to better explain the methodology applied/used/developed in the paper.

We thank the reviewer for their comment. We have expanded our abstract to include more detailed information regarding how changes in personality state were operationalized as well as aspects of the SNA and machine learning components of the study. 

3. A better way to describe subsection 2.2 "Data preprocessing and outcome operationalization" is to include a table and list out the parameters and their ranges. Also, discuss the performance-metrics description (mathematical formulas) used to evaluate the proposed approach.

We appreciate the reviewer’s suggestion. We have added the following table which summarizes the five RMSSD outcome parameters (median, minimum, maximum) across the cohort for each week-based delineation of the log data:

Table 1. Summary of Weekly EMA Data by Personality Item

Week Start Date End Date RMSSD EXTRA RMSSD AGREE RMSSD CONSC RMSSD STABL RMSSD OPEN

 Median [Minimum, Maximum]

1 30-Jan 03-Feb 1.24 [0.33, 3.15] 0.85 [0.13, 1.87] 0.84 [0.24, 2.02] 0.76 [0.22, 2.04] 0.91 [0.22, 2.67]

2 06-Feb 10-Feb 1.11 [0.00, 2.61] 0.85 [0.00, 2.33] 0.73 [0.00, 2.12] 0.79 [0.00, 2.48] 0.97 [0.00, 2.39]

3 13-Feb 17-Feb 1.21 [0.00, 3.47] 0.75 [0.00, 1.99] 0.71 [0.00, 2.22] 0.80 [0.00, 1.78] 0.93 [0.00, 3.11]

4 20-Feb 24-Feb 1.18 [0.20, 2.66] 0.83 [0.00, 2.37] 0.68 [0.00, 1.97] 0.73 [0.00, 2.50] 0.84 [0.00, 3.48]

5 27-Feb 02-Mar 1.04 [0.00, 3.30] 0.80 [0.00, 2.02] 0.75 [0.00, 2.86] 0.66 [0.00, 2.19] 0.93 [0.00, 2.14]

6 05-Mar 09-Mar 1.18 [0.00, 2.42] 0.80 [0.00, 1.89] 0.65 [0.00, 2.05] 0.78 [0.00, 1.75] 0.93 [0.00, 2.30]

In response to reviewer 1, the details for RMSSD (including a formula) has been added to subsection 2.2. Details concerning performance metrics (R2 and RMSE) are now included in the newly reorganized section 2.5 of Methods.

4. There are so many sections with little details. Combine them appropriately and include only the relevant ones.

We thank the reviewer for their comment which has led us to reorganize our Methods section. Specifically, we have combined sections 2.5 - 2.8 into one section entitled “Machine learning modeling and analysis” which has also been expanded upon to address other reviewer comments.

5. The result section needs significant improvement in terms of better discussion regarding the performance analysis of the proposed approach.

We agree with the reviewer that the Results section requires further clarification and contextualization of the various indices presented. To address this deficiency, we have included literature-supported explanations to contextualize the performance of the machine learning models. In applied psychology, a correlation (r) between 0.3 to 0.5 corresponds to a moderate effect or predictive association, and r > 0.5 corresponds to a large effect or predictive association (Rice & Harris, 2005). While we report the R2 values in the present study, which is more common for machine learning studies, taking the square root of these values reveals that, on average, for each of the 5F constructs, large relationships are exhibited (e.g., R2=0.28 for extraversion corresponds to r = 0.53). These associations have been clarified with proper citation in section 3.2 of the Results.

Additionally, the inclusion of the root mean square error (RMSE) metric was used to assess how far off on average the machine learning model predictions were from the true changes in 5F personality states. Because RMSE is context dependent, it is harder to contextualize within the literature; however, the values reported in Table 3 suggest that model predictions exhibited relatively small deviances from the actual outcome values for 52/54 participants. Additional detail pertaining to the interpretation of RMSE as an outcome metric has been included in section 3.2 of the Results.

References:

Rice, M. E., & Harris, G. T. (2005). Comparing effect sizes in follow-up studies: ROC Area,

Cohen’s d, and r. Law and Human Behavior, 29(5), 615–620. 

https://doi.org/10.1007/s10979-005-6832-7

6. The quality of the figures requires improvement. At present many of them are quite blurry.

We have increased the DPI for each of our figures and have increased the font size where possible to improve readability. We have also rearranged and reformatted aspects of Figure 2 for increased clarity.

We greatly appreciate the opportunity to remedy the manuscript’s deficiencies and hope that the manuscript in its new form can be considered for publication in PLoS ONE. We look forward to hearing back from you.

Sincerely,

Manuscript authors

---

## [Decision Letter · Decision Letter 1]

31 Oct 2022

Using passive sensor data to probe associations of social structure with changes in personality: A synthesis of network analysis and machine learning

PONE-D-22-21274R1

Dear Dr. Lekkas,

We’re pleased to inform you that your manuscript has been judged scientifically suitable for publication and will be formally accepted for publication once it meets all outstanding technical requirements.

Kind regards,

Dr. Rajesh Kumar

Academic Editor

PLOS ONE

Additional Editor Comments (optional):

Both reviewers are satisfied with the revised paper.

Reviewers' comments:

Reviewer's Responses to Questions

**Comments to the Author**

1. If the authors have adequately addressed your comments raised in a previous round of review and you feel that this manuscript is now acceptable for publication, you may indicate that here to bypass the “Comments to the Author” section, enter your conflict of interest statement in the “Confidential to Editor” section, and submit your "Accept" recommendation.

Reviewer #1: All comments have been addressed

Reviewer #2: All comments have been addressed

2. Is the manuscript technically sound, and do the data support the conclusions?

Reviewer #1: Yes

Reviewer #2: Partly

3. Has the statistical analysis been performed appropriately and rigorously? 

Reviewer #1: Yes

Reviewer #2: Yes

4. Have the authors made all data underlying the findings in their manuscript fully available?

Reviewer #1: No

Reviewer #2: No

5. Is the manuscript presented in an intelligible fashion and written in standard English?

Reviewer #1: Yes

Reviewer #2: Yes

6. Review Comments to the Author

Reviewer #1: The authors have addressed all my comment with their updates in the manuscript. It can be accepted for publication.

Reviewer #2: All comments and queries are appropriately addressed.Complete data file could be uploaded for readers use.

7. PLOS authors have the option to publish the peer review history of their article (what does this mean?). If published, this will include your full peer review and any attached files.

Reviewer #1: No

Reviewer #2: No

---

## [Editor Report · Acceptance letter]

4 Nov 2022

PONE-D-22-21274R1 

Using passive sensor data to probe associations of social structure with changes in personality: A synthesis of network analysis and machine learning 

Dear Dr. Lekkas:

I'm pleased to inform you that your manuscript has been deemed suitable for publication in PLOS ONE. Congratulations! Your manuscript is now with our production department. 

Kind regards, 

on behalf of

Dr. Rajesh Kumar 

Academic Editor

PLOS ONE